Genetic approaches to the conservation of migratory bats: a study of the eastern red bat (Lasiurus borealis)

Vonhof Maarten J. 1 2 maarten.vonhof@wmich.edu
Russell Amy L. 3
1 Department of Biological Sciences, Western Michigan University , Kalamazoo, MI , USA
2 Environmental and Sustainability Studies Program, Western Michigan University , Kalamazoo, MI , USA
3 Department of Biology, Grand Valley State University , Allendale, MI , USA
Consuegra Sofia
Electronic publication date: 2015 May 28
Publication date: 2015
Volume: 3
Electronic Location ID: e983
Received 2015 Jan 9; Accepted 2015 May 8
Copyright: © 2015 Vonhof and Russell
Copyright year: 2015
Copyright holder: Vonhof and Russell
License: This is an open access article distributed under the terms of the Creative Commons Attribution License, which permits unrestricted use, distribution, reproduction and adaptation in any medium and for any purpose provided that it is properly attributed. For attribution, the original author(s), title, publication source (PeerJ) and either DOI or URL of the article must be cited.
License URL: https://creativecommons.org/licenses/by/4.0/

Keywords: Bats, Coalescent methods, Phylogeography, Migration, Conservation genetics, Effective population size, Wind energy

Funding: U.S. Department of Energy, 20% Wind by 2030 Program DE-EE0000533 We received funding from US Department of Energy, 20% Wind by 2030 Program (grant DE-EE0000533). The funders had no role in study design, data collection and analysis, decision to publish, or preparation of the manuscript.

==============================
Documented fatalities of bats at wind turbines have raised serious concerns about the future impacts of increased wind power development on populations of migratory bat species. However, for most bat species we have no knowledge of the size of populations and their demographic trends, the degree of structuring into discrete subpopulations, and whether different subpopulations use spatially segregated migratory routes. Here, we utilize genetic data from eastern red bats (Lasiurus borealis), one of the species most highly affected by wind power development in North America, to (1) evaluate patterns of population structure across the landscape, (2) estimate effective population size (Ne), and (3) assess signals of growth or decline in population size. Using data on both nuclear and mitochondrial DNA variation, we demonstrate that this species forms a single, panmictic population across their range with no evidence for the historical use of divergent migratory pathways by any portion of the population. Further, using coalescent estimates we estimate that the effective size of this population is in the hundreds of thousands to millions of individuals. The high levels of gene flow and connectivity across the population of eastern red bats indicate that monitoring and management of eastern red bats must integrate information across the range of this species.

Introduction

As concerns about anthropogenic climate change and the long-term environmental impacts of burning of fossil fuels on biological and human systems have heightened, there is increasing motivation to develop alternative sources of energy that will reduce the production of greenhouse gasses. Wind power has become an increasingly important sector of the energy industry and is one of the fastest growing sources of renewable energy (Kaldellis & Zafirakis, 2011; Leung & Yang, 2012). Despite the many positive aspects of wind power development, there have been environmental costs associated with turbine installation and operation (Morrison & Sinclair, 2004; Abbasi et al., 2014). Fatalities of bats at wind power installations have emerged as a major environmental impact of wind power development, with large mortality events being reported at a number of wind energy facilities in the United States and abroad (Erickson et al., 2001; Erickson, Johnson & Young, 2005; Kunz et al., 2007; Arnett et al., 2008). The bat species most affected by wind power in North America are migratory, tree-roosting species such as hoary bats (Lasiurus cinereus), eastern red bats (Lasiurus borealis), and silver-haired bats (Lasionycteris noctivagans), which together constitute almost three-quarters of the bat carcasses found at wind turbines (Arnett et al., 2008). Although mortalities may occur throughout April to November, most bat fatalities in North America have been reported in late summer and early autumn (reviewed by Kunz et al., 2007; Arnett et al., 2008) and appear to be concentrated during fall migration of the affected species (Cryan, 2003).

The observed high levels of mortality for these species at wind power installations raise concerns about the long-term impacts of this technology on bat populations, yet we lack the necessary information to place this mortality in context with respect to baseline population estimates and demographic trends of the affected species. For most bat species we have no knowledge of the size of populations and their demographic trends, the degree of structuring into discrete subpopulations, and whether different subpopulations use spatially segregated migratory routes. While estimates of local population sizes within particular roosts may be feasible using traditional capture-mark-recapture (CMR) methodology or survey techniques, no reliable range-wide population estimates exist for any bat species (O’Shea & Bogan, 2003; Kunz et al., 2009). Traditional demographic approaches have limitations when applied to bats, as they are nocturnal, exhibit cryptic behavior, and are difficult to follow over time during extensive seasonal movements between summer breeding areas and overwintering sites (Cryan, 2003; Rivers, Butlin & Altringham, 2006). The tree-roosting migratory bat species that are killed in high numbers at wind turbines are especially inaccessible for traditional CMR studies, given their solitary nature and restriction to forested habitats (Kunz, 1982; Shump & Shump, 1982a; Shump & Shump, 1982b). Large-scale banding studies typically experience extremely low recapture rates (e.g., Glass, 1982; reviewed in O’Shea & Bogan, 2003), and there are serious data deficiencies with respect to sex- and age-specific survival and reproductive rates that hamper our ability to widely apply demographic models to bat populations. Given these difficulties, we require other approaches to estimating population sizes and demographic trends within migratory bat populations affected by wind power development.

Genetic approaches provide an alternative to traditional demographic methods of population estimation, and allow us to estimate the degree of population structuring, demographic trends within subpopulations, and effective population size (Ne) using data on allele frequencies or the base composition of DNA sequences. Fewer individuals need to be sampled relative to CMR approaches, and individuals need only be sampled a single time for many analyses. In addition, population parameters can be estimated directly from the observed patterns of genetic variation, and age- or sex-specific demographic information may not be required. Molecular markers can also be used to examine levels of population differentiation within a species and to geographically delimit populations or groups of populations based on the observed distribution of genetic variation (Freeland, Petersen & Kirk, 2011). Importantly, such analyses can be used to define the relevant unit for population monitoring, and highlight demographic connections among populations that may not be obvious from behavioral data alone. As mating is likely to take place during migration in bats (Dodd & Adkins, 2007; Cryan, 2008; Cryan et al., 2012; Solick et al., 2012), gene flow should occur among populations that interact during migration. Therefore it is likely that any genetically distinct populations, if they exist, will be using different migratory pathways and may be subject to different mortality rates as wind turbines are concentrated heterogeneously across the landscape. The analysis of genetic population structure is therefore highly relevant to our understanding of bat—wind turbine interactions.

While it is not possible to directly estimate adult census population size (Nc) using molecular data (although genetic markers can be used to identify individuals for traditional CMR analyses; Luikart et al., 2010), it is possible to estimate effective population size (Ne). Ne is defined as the number of individuals in an ideal Wright–Fisher population (a large, constant-sized, randomly-mating, hermaphroditic population with discrete generations) that would lose genetic variation through genetic drift at the same rate as the actual population (Crow & Denniston, 1988). It provides information on how quickly genetic variation is being lost, or relatedness is increasing, in a population of interest, and may be interpreted as an estimate of the number of individuals actually contributing genes to the next generation. The estimation of Ne has seen wide application in studies of threatened or isolated populations, as the magnitude of genetic drift, and hence loss of genetic variation, is inversely proportional to Ne (Leberg, 2005; Wang, 2005; Luikart et al., 2010). Current estimates of Ne can be used to assess the ‘genetic health’ of populations and their capability to respond to future environmental change or anthropogenic changes via selection (Frankham, Ballou & Briscoe, 2002). Estimation of Ne is also common in phylogeographic studies exploring past changes in population sizes in relation to changing climatic conditions or vicariant events in the evolutionary history of species (Avise, 2000; Russell et al., 2011), thus providing important insight into the demographic history of populations and species.

Here, we utilize genetic data from eastern red bats (Lasiurus borealis), one of the species most highly affected by wind power development in North America, to (1) evaluate patterns of population structure and whether different subpopulations use spatially segregated migratory routes, (2) estimate effective population size (Ne), and (3) assess signals of growth or decline in population size. This species was chosen because it is one of the three bat species of greatest concern with regard to the biodiversity impacts of wind energy, and has the highest fatality rate at a number of wind power installations in the eastern United States (Arnett et al., 2008). Although estimates of census population size would be preferable for understanding the size of bat populations and the potential impact of fatalities at wind power installations, Ne estimates may provide us with valuable information on the size of the evolutionarily relevant portion of the population (that portion contributing genes to the next generation). Further, regular monitoring of Ne might serve as a proxy for tracking changes in population size over time. Our study provides valuable data for understanding the population-level impacts of mortalities due to wind power for this migratory bat species by assessing whether there are discrete subpopulations that may represent independent management units and may undergo different migratory behavior, whether populations from different regions may be connected demographically, and the relative magnitude and historical population trends of the population or subpopulations we identify.

Methods

Sampling

Tissue samples from eastern red bats were collected by researchers capturing bats in the field or collecting carcasses at wind power developments. We asked researchers across the range to collect samples, but eastern red bats were not encountered in all areas due to regional differences in encounter rates. Therefore, we have the largest sample sizes per site and the greatest number of samples from the eastern portion of the species’ range. All researchers were required to have appropriate state and federal collecting permits. A small number of samples from Michigan were collected by one of the authors (MJV) under permit from the state of Michigan (Michigan Department of Natural Resources permit SC-1257) with appropriate Institutional Animal Care and Use Committee approval (Western Michigan University protocol 05-03-01).

We compiled a collection of tissue samples from known sample sites collected in the summer months (June to mid-August when bats are likely to be resident) primarily between 2000 and 2006, for the purpose of assessing levels of genetic population structure and estimating Ne (Table 1 and Table S1). We received tissue samples for 1–39 bats from any given site. We had sufficient sample size (N > 15) for each of 12 sites with which to carry out site-level population genetic analyses (Fig. 1 and Table 1). Unlike colonial bats roosting in buildings or trees where bats can be captured in numbers from a single point location during a single sampling session, tree-roosting bats such as eastern red bats are solitary. Sampling these bats therefore must involve the capture of foraging individuals and may encompass individuals from a wider area over a longer time scale. Therefore we define a ‘site’ as a collection of capture localities within a set of nearby counties within a single state or province. For six of our sites, bats were captured either within a single county or at a single capture location (AR, GA, MO, ON, TX, WV-Ma), while the other six sites consisted of individuals captured in several counties within a given state (IL, MD, MI, NC, TN, WV-Pe; the site label for the latter site represents one of the two counties included; Table S1). There were no consistent differences in diversity measures within sites or levels of differentiation between sites associated with sites containing samples from a single versus multiple counties (see ‘Results’).

Figure 1 Map showing the range of eastern red bats and all sampling locations.

Only labeled locations (black dots) had sufficient sample sizes to be included in population-level analyses, and labels reflect two-letter state or province codes (two sampling locations within West Virginia are further labeled with the first two letters of the county to distinguish them). The range map source is the IUCN (http://www.iucnredlist.org/details/11347/0): Arroyo-Cabrales J, Miller B, Reid F, Cuarón AD, de Grammont PC. 2008. Lasiurus borealis. In IUCN 2014. IUCN Red List of Threatened Species. Version 2014.2. < www.iucnredlist.org>. Downloaded on November 5, 2013.

Table 1 Sites sampled and diversity statistics for 16-locus microsatellite genotypes and mitochondrial HV2 sequences.

Site labels represent two-letter state codes as in Fig. 1.

Site	State or province	N Gen	NA	HO	HE	AR	ARPriv	F IS	N Seq	NH	h	π	
AR	Arkansas	39	18.25	0.84	0.88	13.14	0.50	0.044	25	21	0.987	0.016	
GA	Georgia	30	16.75	0.81	0.87	13.12	1.16	0.064	17	13	0.963	0.009	
IL	Illinois	26	15.31	0.80	0.87	12.88	0.56	0.084	26	22	0.985	0.013	
MD	Maryland	21	13.31	0.81	0.86	12.19	0.80	0.057	15	15	1.000	0.012	
MI	Michigan	17	12.69	0.82	0.88	12.69	0.84	0.073	16	16	1.000	0.013	
MO	Missouri	27	16.25	0.84	0.89	13.20	0.80	0.056	34	21	0.961	0.009	
NC	North Carolina	18	13.19	0.81	0.88	12.87	0.76	0.079					
ON	Ontario	19	14.13	0.87	0.88	13.43	1.05	0.021	19	17	0.983	0.012	
TN	Tennessee	22	14.50	0.82	0.87	12.98	0.79	0.065	26	23	0.991	0.010	
TX	Texas	20	14.19	0.79	0.88	13.14	0.81	0.105	21	20	0.995	0.011	
WV-Pe	West Virginia	20	13.25	0.83	0.87	12.35	0.79	0.050	19	18	0.994	0.010	
WV-Ma	West Virginia	25	15.44	0.85	0.88	13.02	0.45	0.036					
Overall		284	14.77	0.82	0.88	12.92	0.78	0.061	218	18.6	0.986	0.011	
Notes.

NGen number of individuals genotyped

NA number of alleles

HO observed heterozygosity

HE expected heterozygosity

AR allelic richness

ARPriv private allelic richness

FIS inbreeding coefficient

NSeq number of individuals sequenced at mitochondrial HV2 locus

NH number of haplotypes

h haplotype diversity

π nucleotide diversity

Overall values represent means for all measures except NGen and NSeq, which represent sums.

Laboratory methods

Analyses of population genetic structure were carried out by analyzing variation at microsatellite loci and mtDNA sequences. Ne estimation was carried out using these same markers, as well as sequences from a nuclear intron marker. All but one analysis (msvar; see below) used this primary set of marker data.

DNA was extracted from samples using a DNEasy Tissue Extraction Kit (Qiagen, Hilden, Germany). Sixteen variable microsatellite loci were genotyped for all individuals used in site-level analyses (N = 284) using primers developed specifically for eastern red bats (primers Lbo-B06, C07, D08, D200, D202, D203, D204, D226, D240, D245, and D248; M Eackles & T King, pers. comm., 2011), as well as primers originally developed for other bat species (MS3E10 and MS1C01, Trujillo & Amelon, 2009; IBat-Ca22, Oyler-McCance & Fike, 2011; Cora_F11_C04, Piaggio, Figueroa & Perkins, 2009; and Coto_G12F_B11R, Piaggio et al., 2009). Loci were multiplexed whenever possible; all PCR reactions combined varying amounts of each primer and 2 µL template DNA with an illustra PuReTaq ready-to-go PCR bead (GE Health Care) to a total volume of 25 µL (Table S2). The basic cycling conditions consisted of 1 min at 94 °C, three cycles of 30 s at 94 °C, 20 s at Ta (54 or 60 °C), and 5 s at 72 °C, 33 cycles of 15 s at 94 °C, 20 s at Ta, and 10 s at 72 °C, followed by a final extension at 72 °C for 30 min. Some amplifications required additional cycles or the removal of the final extension step (Table S2). Multiple PCR reactions were subsequently pooled for loading on an ABI3130 Sequencer at the Vanderbilt University DNA Sequencing Facility for fragment analysis (see Table S2 for information on multiplexes and loads used), and visualized and scored using GeneMarker software (SoftGenetics, State College, Pennsylvania, USA).

A fragment of the hypervariable 2 portion of the mitochondrial DNA control region (hereafter HV2) was sequenced from 218 individuals used in site-level analyses (because of financial constraints not all individuals from each location and not all locations were sequenced; Table 1), as well as 77 bats from 30 additional locations that were not included in site-level analyses, for a total of 295 individuals sequenced. Amplification of HV2 was initially carried out using the reverse complement of primer F from Wilkinson & Chapman (1991; RevF: 5′-CTA CCT CCG TGA AAC CAG CAA C-3′) sitting in the central conserved sequence block as the forward primer, and the primer sH651 located in the tRNAPro gene (Castella, Ruedi & Excoffier, 2001) as the reverse primer. However, these primers span a region containing a large stretch of 6 bp repeats, resulting in a large amplicon of 1,500–2,000 bp. We therefore designed a new reverse primer (LABO-HV2R2: 5′-TCC TGT WAC CAT TAA YTA ATA TGT CCC-3′) that amplified a 408 bp fragment excluding the repeats. Amplification was carried out using the above reaction conditions and the cycling conditions in Castella, Ruedi & Excoffier (2001) with a Ta of 60 °C. PCR reactions were cleaned using ExoSAP-IT (PCR Product Pre-Sequencing Kit; Affymetrix, Santa Clara, California, USA), and sent to the University of Arizona Genetics Core for bi-directional sequencing. Sequences were edited using CodonCode Aligner software (CodonCode Corporation, Centerville, Massachusetts, USA). All unique HV2 haplotypes are deposited in Genbank (accession numbers KR337091–KR337257; Table S1).

We further sequenced a 612 bp fragment of the nuclear Chymase intron 4 (CHY) for a random subset of 88 individuals. Based on our results indicating panmixia across the sampled range of eastern red bats (see ‘Results’), a random sample of individuals should represent genetic variation found in the wider population (Felsenstein, 2006). This reduced subsample was chosen because the methods used for Ne estimation are computationally intensive, and analysis would not have been possible with a larger sample of sequences. CHY was amplified through PCR using the primers Chy-F (5′-GTC CCA CCT GGG AGA ATG TG-3′) and Chy-R (5′-TGG GAG ATT CGG GTG AAG-3′; Venta et al., 1996). The reaction conditions were identical to those for the microsatellite loci, except that the reaction used just 1 µL of template. The temperature profile included an initial extended denaturation of 95 °C for 5 min, followed by 40 cycles of 95 °C for 1 min, 52 °C for 1 min and 72 °C for 1.5 min, with a final extension step at 72 °C for 4 min. The PCR reaction was cleaned using a PCR purification kit (Qiagen, Hilden, Germany) and sent to the University of Arizona Genetics Core for bi-directional sequencing using the Chy-F and Chy-R primers. These diplotypes were edited and heterozygous sites called using Sequencher v.4.8 (GeneCodes).

Some individuals (N = 36) found to contain two or more heterozygous sites were cloned using the TOPO TA cloning kit (Life Technologies, Carlsbad, California, USA) following manufacturer’s instructions. Six to eight colonies were picked for each cloned individual. The picked colonies were each suspended in 10 µL dH2O and heated to 95 °C for 10 min to lyse the cells. The cell lysate was then used directly as template DNA for colony screening through PCR. The PCR reaction combined 10 ng of each primer and 10 µL cell lysate with an illustra PuReTaq ready-to-go PCR bead (GE Health Care, Little Chalfont, UK) to a total volume of 25 µL. The temperature profile followed that described above for the initial cloned PCR. PCR reactions yielding amplicons of the expected size (∼650 bp) were cleaned using ExoSAP-IT (Affymetrix, Santa Clara, California, USA) following the manufacturer’s instructions. Cleaned PCR amplicons were then sent to the University of Arizona Genetics Core for bi-directional sequencing using the Chy-F and Chy-R primers. Based on these experimentally-resolved haplotypes, another 44 individuals with ambiguous diplotypes were computationally phased using Phase v.2.1.1 (Stephens, Smith & Donnelly, 2001; Stephens & Donnelly, 2003) with a confidence threshold of 0.95. All unique CHY haplotypes are deposited in Genbank (accession numbers KR362302–KR362477; Table S1).

Analysis of genetic structure

For microsatellite genotypes, deviations from Hardy–Weinberg equilibrium (HWE) at each locus were estimated using GENODIVE (Meirmans, 2012), and loci were confirmed to be in linkage equilibrium using FSTAT v.2.9.3 (Goudet, 1995). Null allele frequencies for each locus were estimated in CERVUS v.3.1 (Kalinowski, Taper & Marshall, 2007). To test for differences among sites in levels of genetic diversity, several indices of nuclear genetic diversity were estimated, including number of alleles per locus, allelic richness, and the inbreeding coefficient (FIS) using FSTAT, private allelic richness using HP-RARE 1.0 (Kalinowski, 2005), and observed and expected heterozygosity using GENODIVE. We then tested for differences among sites (or groups of sites) in allelic richness, and FIS in FSTAT, and expected heterozygosity in GENODIVE, using 10,000 permutations.

Different clustering algorithms can produce different solutions, and concordance among multiple techniques is suggestive of the presence of a strong genetic signal (Guillot et al., 2009). Therefore, we applied two different approaches to determine the most likely number of distinct genetic clusters independent of original sampling locations. First, we utilized the model-based Bayesian clustering approach in STRUCTURE v.2.3.3 software (Pritchard, Stephens & Donnelly, 2000; Falush, Stephens & Pritchard, 2003) with population membership as a prior (Hubisz et al., 2009). To determine the optimal number of clusters (K), we ran 10 runs per K, for K = 1–10, each with an MCMC search consisting of an initial 100,000-step burn-in followed by 400,000 steps using the admixture model with correlated allele frequencies. The most likely number of clusters was determined using the Evanno, Regnaut & Goudet (2005) method implemented in the program STRUCTURE HARVESTER (Earl & vonHoldt, 2012). The Evanno, Regnaut & Goudet (2005) method is not informative for the highest and lowest K values; therefore, if the highest log likelihood value was observed for K = 1 or 10 across all replicates, we accepted that as the best-supported value of K.

Second, we applied the repeated allocation approach of Duchesne & Turgeon (2009) and Duchesne & Turgeon (2012) implemented in the software FLOCK. In this method, samples are initially randomly partitioned into K clusters (K ≥ 2), allele frequencies are estimated for each of the K clusters, and each genotype is then reallocated to the cluster that maximizes the likelihood score. Repeated reallocation based on likelihood scores (20 iterations per run) results in genetically homogeneous clusters within a run (Duchesne & Turgeon, 2012). Fifty runs were carried out for each K, and at the end of each run the software calculated the log likelihood difference (LLOD) score for each genotype (the difference between the log likelihood of the most likely cluster for the genotype and that of its second most likely cluster) and the mean LLOD over all genotypes. Strong consistency among runs (resulting in ‘plateaus’ of identical mean LLOD scores) is used to indicate the most likely number of clusters (Duchesne & Turgeon, 2012).

The level of genetic differentiation among pre-defined sites (N = 12; Table 1) based on microsatellites was determined by calculating pairwise distance measures, including FST (Weir & Cockerham, 1984) in ARLEQUIN v.3.11 (Excoffier, Laval & Schneider, 2005), and a measure independent of the amount of within-site diversity (Jost’s D; Jost, 2008) in GENODIVE. We tested for significance of pairwise FST values between sites with 10,000 permutations, and performed an analysis of molecular variance (AMOVA; Excoffier, Smouse & Quattro, 1992) to describe the relative amount of genetic variation within and among sites in ARLEQUIN.

To describe overall levels of mtDNA diversity within sites, we calculated haplotype (h) and nucleotide (π) diversities in DnaSP v.5.10.1 (Librado & Rozas, 2009). We calculated pairwise FST values between sites and tested for significance with 10,000 permutations in ARLEQUIN to identify pairs that were genetically distinct. As with microsatellite genotypes, we performed an AMOVA on HV2 haplotype frequencies in ARLEQUIN.

Estimation of Ne

We used a number of approaches to estimate Ne for eastern red bats. Although we originally set out to estimate the short-term variance effective population size (NeV, Crandall, Posada & Vasco, 1999), it quickly became apparent that Ne was very large (see ‘Results’). This constraint precluded the use of single sample estimators based on linkage disequilibrium or summary statistics (Waples & Do, 2009; Waples & Do, 2010; Tallmon, Luikart & Beaumont, 2004; Tallmon et al., 2008), which are only effective for Ne < 1,000, or temporal methods (e.g., Jorde & Ryman, 1995), which are based on changes in allele frequencies due to genetic drift between time points (as drift is negligible with large Ne). Furthermore, the cohort-based demographic data required for the Jorde & Ryman (1995) method were simply not available for any bat species.

Therefore, we focused on coalescent analyses, using three primary methods to estimate long-term inbreeding effective population size (NeI, Crandall, Posada & Vasco, 1999). These methods utilize different types of data, and therefore provide complementary estimates based on differences in the mutation rates of the markers used and differences in the underlying models assumed.

IMa2

We used the coalescent-based software IMa2 (release date 27 August 2012; Hey, 2010a; Hey, 2010b) to estimate the effective size of the panmictic eastern red bat population (see ‘Results’). The analysis included the CHY and HV2 sequences and 16-locus microsatellite genotypes. One hundred microsatellite genotypes (=200 chromosomes) for each locus were subsampled at random out of the full dataset in order to reduce the computational time of the analysis. The DNA sequence data (CHY and HV2) were edited to conform to an infinite sites model of mutation; microsatellite data were analyzed assuming a single-step model of mutation.

In the IMa2 analysis, we modified the underlying population model to consider only a single population, with a uniform prior on the size of that population varying from θ = 0.05 to 99.95. We ran 40 heated chains for an initial burn-in of ∼3.6 million steps, followed by an MCMC search of ∼10.2 million steps. Stationarity of the search chains was validated by monitoring ESS values.

Lamarc

We used the software package Lamarc v.2.1.8 (Kuhner, 2006) to estimate effective population size and population growth rates independently for the nuclear CHY and the mitochondrial HV2 sequence data. We considered a model of a single panmictic population that undergoes population size change (growth or decline) until it reaches the current population size. We implemented a Bayesian analysis in Lamarc with priors on θ ranging from 10−5 to 50 and on the population size change parameter (g) ranging from −500 to 2,000. The data were analyzed in three independent runs, with each run consisting of an MCMC search that was 20 million steps long and sampled every 200 steps. The first 2 million steps were discarded as a burn-in. Each MCMC search was run as 3 heated chains, with relative heating temperatures of 1, 1.5, and 3, and each search was replicated three times internally within each of the independent runs. Posterior distributions for each independent run and for overall results per locus were visualized using Tracer v.1.5. Results are reported as median point estimates with 95% confidence intervals. All parameter estimates were well supported, with ESS values exceeding 100 in all cases. Ne was calculated from the estimated coalescent-scaled parameter θ using the equations: θ = Neμ for mitochondrial data and θ = 4Neμ for autosomal data, where Ne is the effective size of the entire population. This software uses mutation rates in units of substitutions per site per generation; based on the relative mutation rates estimated for the same data in the IMa2 analysis, we used a mutation rate of 4.29 × 10−8 per site per generation for the HV2 dataset and 7.76 × 10−9 per site per generation for the CHY dataset.

msvar

The third approach we used was the coalescent-based software msvar v.1.3 (Beaumont, 1999), which estimates effective population size and demographic trends from microsatellite genotype data. This analysis considers a model in which a single ancestral population of size NA experiences exponential population size change beginning at time t until the population reaches the current size N1. Unlike IMa2 and Lamarc, which calculate only long-term average Ne, msvar separately calculates current and ancestral Ne. Therefore, rather than use the microsatellite genotypes included in all other site-level analyses (which spanned a multi-year period), we generated microsatellite genotypes following the methods outlined above for two specific years for which we had sufficiently large sample size (2002: N = 353 and 2010: N = 226). These datasets were analyzed separately to determine whether mortality over that time interval had a measurable effect on estimates of Ne. Samples of genotypes for 2002 and 2010 were each comprised of a mixture of individuals of known summer origin, as well as bats of unknown origin killed at wind power developments during fall migration.

To make the msvar analysis computationally feasible, we randomly subsampled 100 diploid individuals from each time point (2002 and 2010). Subsampling was performed twice, producing subsamples A and B for each time point, to ensure that no bias was introduced through subsampling. Each analyzed dataset thus included 100 sixteen-locus genotypes (=200 chromosomes) from a single year (2002 or 2010).

The msvar analysis requires the specification of hyperpriors for each of the four parameters, N1, NA, t, and the mutation rate μ. These hyperpriors describe distributions from which the locus-specific initial parameter values are drawn, and are given here as [log10(N1), log10(NA), log10(μ), log10(t)]. The parameter means were assumed to be normally distributed with means (7, 7, −3.5, 4.3) and standard deviations (3.5, 4, 0.5, 2). We chose these values for (1) N1 based on estimates of Ne for eastern red bats from our own Lamarc analyses with a relatively large standard deviation to reflect our own uncertainty regarding this parameter, (2) NA based on a null hypothesis of no change in population size with a larger standard deviation to accommodate increased uncertainty in historical parameters, (3) µ based on Storz & Beaumont’s (2002) msvar analysis of microsatellite variation in Cynopterus fruit bats, and (4) t based on a hypothesis of population size change associated with the Last Glacial Maximum with a relatively large standard deviation to reflect our own uncertainty regarding this parameter. The parameter standard deviations were assumed to be normally distributed with means (0, 0, 0, 0) and standard deviations (0.5, 0.5, 2, 0.5). The means of the parameter standard deviations were set to 0 to start the search algorithm with no inter-locus variation; the standard deviations of the parameter standard deviations followed recommendations of Storz & Beaumont (2002). Each of the four datasets (2 time points, with 2 subsamples each) were analyzed 2–3 times, with each run lasting ∼750 million to 2 billion steps and output logged every 100,000 steps. The initial 10% of the MCMC chains from each run were excluded as a burn-in.

Results

Genetic structure

All microsatellite loci were unlinked and the majority of loci met HWE expectations in most populations. MS3E10 was out of HWE in 2 of 12 sites (MO, ON), IBat Ca22 in 2 sites (GA, IL), LboD202 in one site (AR), LboD204 in one site (WV-Pe), and LboD226 in 3 sites (GA, MI, WV-Ma). Mean observed and expected heterozygosities within sites were high (0.82 and 0.88, respectively), as was the mean number of alleles per locus (14.77) and allelic richness (12.92), although private allelic richness was low (0.78; Table 1), and there were no significant differences among sites in allelic richness, FIS, or expected heterozygosity (P > 0.05 in all cases). Diversity statistics per locus are presented in Table S3. Null allele frequencies per locus were generally low and <0.1, except for locus LboD226 with a frequency of 0.123 (Table S3). FST estimates with null alleles are unbiased in the absence of population structure (Chapuis & Estoup, 2007), and removing loci that failed to meet HWE in some sites from the analyses made no difference in our conclusions; therefore we present analyses with all loci included.

AMOVA analysis of microsatellite genotypes indicated an almost complete lack of structure (FST = 0.0044, P < 0.001), with pairwise FST and Jost’s D values between populations consistently low and non-significant (Table 2; FST range: −0.005–0.009; Jost’s D range: −0.036–0.068). Log likelihood values for K = 1 and K = 2 in the Bayesian clustering method (STRUCTURE) were nearly identical (Table S4), and there was no basis upon which to conclude that the most likely number of clusters was different from K = 1 given the low FST values among all sampled sites. Similarly, the repeated reallocation clustering method (FLOCK) failed to reach a plateau for any K > 1, indicating K = 1 as the most likely number of genetic clusters.

Table 2 Pairwise FST (below diagonal) and Jost’s D (above diagonal) values based on 16-locus microsatellite genotypes.

No pairwise FST values were significant based on 10,000 permutations.

Site	AR	GA	IL	MD	MI	MO	NC	ON	TN	TX	WV-Pe	WV-Ma	
AR	–	0.027	0.02	0.001	0.013	0.018	0.001	−0.025	0.025	0.033	0.025	0.001	
GA	0.004	–	0.041	0.02	0.039	0.068	0.011	0.027	0.029	0.047	0.054	0.029	
IL	0.003	0.006	–	0.026	0.046	0.037	0.011	0.007	0.018	0.055	0.037	0.026	
MD	0	0.003	0.004	–	0.012	0.022	0.035	0.009	0.012	0.041	0.02	−0.022	
MI	0.002	0.006	0.007	0.002	–	0.001	−0.033	0.012	0.015	0.012	0.021	0.026	
MO	0.003	0.009	0.005	0.003	0	–	0.006	0.015	0.052	0.049	0.013	0.006	
NC	0	0.002	0.002	0.005	−0.004	0.001	–	−0.017	0.02	−0.002	0.018	0.003	
ON	−0.003	0.004	0.001	0.001	0.002	0.002	−0.002	–	0.026	−0.036	0.024	0.023	
TN	0.004	0.004	0.003	0.002	0.002	0.007	0.003	0.004	–	0.031	0.029	0.026	
TX	0.005	0.007	0.008	0.006	0.002	0.006	0	−0.005	0.004	–	0.021	0.039	
WV-Pe	0.004	0.008	0.005	0.003	0.003	0.002	0.002	0.003	0.004	0.003	–	0.011	
WV-Ma	0	0.004	0.004	−0.003	0.004	0.001	0.001	0.003	0.004	0.006	0.002	–	

We observed 167 unique haplotypes representing 84 segregating sites among the 295 individuals sequenced at the mitochondrial HV2 locus. The number of haplotypes per site ranged from 13 to 23 (mean = 18.6), and haplotype diversity (h, mean = 0.986, range = 0.961–1) was high for all sites (Table 1). However, nucleotide diversity (π, mean = 0.011, range = 0.009–0.016) was relatively low for all sites (Table 1). AMOVA analysis indicated very low levels of mitochondrial differentiation among sites (FST = 0.0113, P < 0.05; 1.13% of the variation is explained by differences among sampling sites, and 98.87% of the variation occurs within sites). Accordingly, pairwise FST values among sites were consistently low and ranged from −0.03 to 0.049 (Table 3), with only two significant values (between the IL and MO and the IL and TX sites).

Table 3 Pairwise FST values based on mitochondrial HV2 sequence data.

Site	AR	GA	IL	MD	MI	MO	ON	TN	TX	
AR	–									
GA	0.012	–								
IL	−0.008	0.032	–							
MD	0.014	0.037	0.019	–						
MI	−0.006	−0.006	0.006	−0.011	–					
MO	0.024	0.021	0.037*	0.032	0.016	–				
ON	0.005	0.014	−0.005	0.008	0.000	−0.006	–			
TN	0.006	0.009	0.001	0.030	0.005	0.008	−0.004	–		
TX	0.028	0.049	0.042*	0.021	0.020	0.013	0.000	0.036	–	
WV-Pe	0.003	0.015	−0.001	0.014	0.005	−0.009	−0.030	−0.016	0.009	
Notes.

* Significant values based on 10,000 permutations (P < 0.05).

Ne estimation

We used three coalescent methods to estimate Ne for eastern red bats: IMa2, Lamarc, and msvar. These methods utilize different suites of data (microsatellites only for msvar, nuclear and mitochondrial sequence data only for Lamarc, all three data types for IMa2), and therefore were expected to provide complementary estimates based on differences in the mutation rates of the markers used and differences in the underlying models assumed.

IMa2

This analysis converged on an unambiguous, unimodal posterior distribution for the single population parameter θ (=4Neμ) for the panmictic eastern red bat population. The most probable value of θ was estimated to be 37.95 (95% CI [32.15–45.55]). We used Pesole et al.’s (1999) estimate of mammalian mitochondrial mutation rates (=2.740 × 10−8 substitutions per site per year) to calculate locus-specific mutation rates for our data. The geometric mean of these rates (=8.03 × 10−6 substitutions per locus per year =1.61 × 10−5 substitutions per locus per generation; Table S5) was used to convert coalescent-scaled estimates of θ into estimates of Ne. Our analysis thus supports an effective population size of approximately 5.91 × 105 individuals (95% CI [5.00–7.09] ×105; Fig. 3).

Lamarc

We used coalescent-based analyses in Lamarc to provide estimates of θ and population growth independently for the nuclear CHY and mitochondrial HV2 loci. Analyses of both markers provided unambiguous, unimodal posterior probability distributions for both parameters. Utilizing the relative mutation rates estimated from IMa2, estimates of Ne using HV2 across three runs in Lamarc were 5.18 × 105 (95% CI [4.25–7.22] ×105; Table 4). The estimate of Ne using CHY (males and females) was significantly larger, with a mean of 1.52 × 106 (95% CI [1.05–2.18] ×106; Table 4). There was a clear signal of historical population growth recovered from both loci (Table 4); however, the time scale over which this growth occurred is not estimated in the Lamarc model.

Table 4 Estimates of θ, Ne, and population growth (g) based on Lamarc analyses.

	θ (95% CI)	Ne (95% CI)	g (95% CI)	
HV2	
Run 1	0.022	5.0 × 105	964.25	
	(0.018, 0.031)	(4.16–7.26 × 105)	(361.03, 1007.18)	
Run 2	0.024	5.52 × 105	965.75	
	(0.019, 0.029)	(4.33–6.78 × 105)	(358.34, 1007.50)	
Run 3	0.022	5.0 × 105	965.95	
	(0.018, 0.033)	(4.25–7.61 × 105)	(382.04, 1006.35)	
Overall	0.022	5.18 × 105	965.32	
	(0.018, 0.031)	(4.25–7.22 × 105)	(367.14, 1007.01)	
CHY	
Run 1	0.048	1.54 × 106	958.85	
	(0.033, 0.067)	(1.07–2.15 × 106)	(496.01, 1002.27)	
Run 2	0.046	1.50 × 106	957.10	
	(0.032, 0.067)	(1.03–2.17 × 106)	(486.19, 1002.01)	
Run 3	0.047	1.52 × 106	952.73	
	(0.033, 0.069)	(1.06–2.21 × 106)	(479.76, 1001.04)	
Overall	0.047	1.52 × 106	956.23	
	(0.033, 0.068)	(1.05–2.18 × 106)	(487.32, 1001.77)	

msvar

Although we found considerable variation from run to run, there were some clear patterns that emerged from these analyses. Importantly, we found no consistent difference between parameter estimates from the 2002 vs. 2010 time points (Fig. 2 and Figs. S1–S2). We also found no consistent difference between independent subsamples (A vs. B, each run 2–3 times) of the full dataset (runs A1–A3 vs. B1 and B3 for 2002; runs A1–A3 vs. B1–B3 for 2010). For the current effective population size N1, we recovered generally consistent estimates on the order of 104–105 (average N1 ≈ 74,500). Estimates of ancestral effective population size NA were less consistent among runs, but did result in estimates ranging in the same order of magnitude as N1 (average NA ≈ 194,300; Fig. S1). These analyses yielded differing signals of population growth vs. decline between runs (Table 5), although a majority of runs (8 of 11) support a model of population decline rather than growth. The time of this population size change (t) was also variable among runs, but generally was on the order of 103–104 years (average t ≈ 21,600 years; Table 5 and Fig. S2). While the time of population size change is difficult to pinpoint with great accuracy, these analyses clearly are not informative regarding very recent population size change.

Figure 2 Tukey boxplot of current Ne from msvar analyses.

Estimates are given on the log10 scale. Datasets A and B represent different subsamples of the full dataset from each respective year.

Figure 3 Posterior probability of Ne for eastern red bats, estimated using IMa2.

The analysis includes autosomal DNA sequence data, mitochondrial DNA sequence data, and autosomal microsatellite genotype data.

Table 5 Estimates of current and ancestral Ne, time of growth and population trend based on msvar analyses.

Year	Current Ne (mode ± variance)	Ancestral Ne (mode ± variance)	Time of growth (mode ± variance)	Trend	
2002_A1	125,786 ± 4.9	24,191 ± 4.5	5,353 ± 1.9	Growth	
2002_A2	21,120 ± 6.2	57,497 ± 3.1	6,924 ± 1.9	Decline	
2002_A3	106,925 ± 3.3	22,460 ± 6.0	27,256 ± 5.8	Growth	
2002_B1	137,848 ± 6.1	14,626 ± 4.6	11,710 ± 3.3	Growth	
2002_B3	195,164 ± 4.4	651,754 ± 3.1	21,915 ± 1.1	Decline	
2010_A1	46,279 ± 3.5	59,872 ± 2.6	88,776 ± 1.9	Decline	
2010_A2	36,766 ± 5.6	427,688 ± 3.8	16,088 ± 4.3	Decline	
2010_A3	24,733 ± 2.4	44,036 ± 5.5	32,866 ± 1.6	Decline	
2010_B1	22,845 ± 3.8	81,332 ± 7.6	5,161 ± 4.2	Decline	
2010_B2	12,670 ± 5.4	29,191 ± 3.8	9,978 ± 1.3	Decline	
2010_B3	89,050 ± 10.0	724,656 ± 2.8	11,552 ± 7.2	Decline	

Discussion

We observed extremely low levels of population structure and effective panmixia across the sampled sites for eastern red bats using both nuclear and mitochondrial DNA markers. Furthermore, there is no evidence for the historical use of different migratory pathways and no evidence for any barriers to gene flow among any of the sampled localities. Few geographic barriers to the movement of vagile organisms such as bats exist east of the Rocky Mountains, and therefore there are likely few impediments to the movement of individuals across the landscape. Phylogeographic studies of widespread bats and birds have shown low levels of genetic differentiation among eastern North American populations (Gibbs, Dawson & Hobson, 2000; Kimura et al., 2002; Jones et al., 2005; Turmelle, Kunz & Sorenson, 2011; Irwin, Irwin & Smith, 2011; but see Miller-Butterworth et al., 2014). When present, genetic structure in these species is often restricted to broad-scale differentiation between eastern and western populations on either side of the Rocky Mountains. In the case of eastern red bats, evidence from museum records indicates that they most likely migrate from northern parts of their range to the southeastern United States (Cryan, 2003) where they roost in trees during warmer periods and may hibernate beneath leaf litter for short durations during colder temperatures (Saugey et al., 1998; Moorman et al., 1999; Mormann & Robbins, 2007). However, there are summer resident populations in the southeastern United States that likely do not migrate, and it is possible that there is variation in migratory tendency across the range of eastern red bats, much like tricolored bats (Perimyotis subflavus; Fraser et al., 2012). Mating likely takes place before or during migration in eastern red bats (Dodd & Adkins, 2007; Cryan, 2008; Cryan et al., 2012; Solick et al., 2012), and can take place before bats hibernate or during warm periods on the wintering grounds. Thus, the potential for mating, and hence gene flow, among individuals that spent their summers in geographically disparate areas during migration or on the wintering grounds is likely very high.

In most colonial temperate bat species, females are philopatric to natal nursery colonies or undergo short dispersal distances to nearby colonies while mating takes place during swarming and/or hibernation at distant sites that act as hotspots of gene flow between bats occupying distant roosts during the summer (Kerth et al., 2003; Veith et al., 2004; Furmankiewicz & Altringham, 2007). As a consequence, levels of mitochondrial differentiation (indicative of female movements) are often quite high among summer maternity colonies while levels of nuclear differentiation (indicative of gene flow through mating) are typically low (Castella, Ruedi & Excoffier, 2001; Bilgin et al., 2008; Kerth et al., 2008; Vonhof, Strobeck & Fenton, 2008; Bryja et al., 2009; Lack, Wilkinson & van den Bussche, 2010; Turmelle, Kunz & Sorenson, 2011). Eastern red bats and other members of the genus Lasiurus roost solitarily in foliage during the summer (Shump & Shump, 1982a; Shump & Shump, 1982b), and if they exhibited philopatry it would likely occur within broader landscape units such as forest patches or stands rather than a single roost. The absence of significant mitochondrial differentiation among samples of eastern red bats suggests that females may be exhibiting high levels of dispersal, and that gene flow likely takes place via both male and female movements and mating (e.g., Russell, Medellín & McCracken, 2005; Vonhof, Strobeck & Fenton, 2008).

Before undertaking our study, we had no prior knowledge of whether the eastern red bat was divided into a series of discrete subpopulations, possibly undertaking migration along different pathways and possibly varying in size, or whether it functioned as a single, panmictic population of unknown size. Our estimates of Ne varied considerably (over an order of magnitude) among the different approaches we used, ranging from 7.45 × 104 based on microsatellite genotypes only (msvar), to 1.52 × 106 for sequence data only (CHY in Lamarc), with intermediate estimates of 5.18 × 105 for HV2 (Lamarc) and 5.91 × 105 using all markers combined (IMa2). This variation is the result of methodological differences among the approaches we used, which all utilize different aspects of the data and make varying assumptions about the underlying historical population processes that may have occurred. Further, the analyses each used different marker data, which vary in their mutation rates, and so are providing estimates across varying time scales. Nevertheless, in combination with the results of population structure analyses, our data indicate that eastern red bats form a single, large, panmictic population across their range and that minimum effective population sizes are likely in the hundreds of thousands.

The parameter most relevant to management of this species, the actual number of individuals in the population (Nc), is not obtainable from our estimates of Ne. A variety of factors may reduce Ne relative to Nc, including fluctuations in population size over time, overlapping generations, and variation among individuals in reproductive success. Attempts have been made to compare estimates of Ne to Nc, and across a wide range of organisms the average Ne/Nc ratio is 0.11–0.14 (Frankham, 1995; Palstra & Ruzzante, 2008); for mammals alone, the average ratio is 0.34 (Frankham, 1995). If we applied this latter mean ratio (0.34) to our point estimates of Ne, we would obtain Nc estimates of 2.19 × 105 to 4.5 × 106 individuals. However, there are a number of serious problems with the use of our coalescent estimates in this way. Ne is a theoretical concept that relates the genetic characteristics of a population to those expected of an ideal population under a Wright–Fisher model. We can evaluate Ne as a measure of the evolutionary potential of populations, but there is no clear relationship between current demography and changes in genetic variation that influence coalescent estimates of Ne. Further, there are a number of methodological concerns. First, Ne has most often been estimated for very small populations of less than 1,000 individuals, and we do not know how the Ne/Nc ratio may vary with the magnitude of Nc. Second, the majority of the ratios provided by Frankham (1995) utilize demographic, rather then genetic, estimates of Ne, and demographic estimates may differ substantially from genetic estimates even when population sizes are small (Luikart et al., 2010). Third, the majority of estimates in Frankham (1995) come from organisms with very different life histories than bats, and we do not know to what extent the Ne/Nc ratio might vary from the overall mean for bats (or most other organisms). Fourth, the calculation of Ne using coalescent-based methods requires division of estimates of θ by the mutation rate (μ) to obtain values of Ne, but mutation rates are extremely difficult to estimate and few good estimates exist for any gene (Ho et al., 2006; Montooth & Rand, 2008; Nabholz, Glémin & Galtier, 2009), much less for any bat species. As a result, any inaccuracy in the mutation rate estimate is amplified arithmetically in the subsequent calculation of Ne (Ovenden et al., 2007; Luikart et al., 2010). Therefore, applying a standard conversion to convert Ne to Nc is highly problematic, and it is best to use our estimates to indicate relative orders of magnitude of bat population sizes rather than to provide any specific population size estimates.

The potential value of our estimates of Ne is that they may be used as a baseline for future monitoring. Assuming fatality rates at wind turbines remain high and continue to grow as wind energy development continues, it is possible that regular estimates of Ne could be utilized to document population trends of affected species (Antao, Perez-Figueroa & Luikart, 2011). Regional projections of bat fatalities predict annual fatality rates numbering in the tens of thousands (Kunz et al., 2007), and the total number of fatalities is likely to continue to rise as wind power development expands. However, the loss of genetic variation from populations and declines in Ne estimates based on linkage disequilibrium are only apparent when population sizes are very small (e.g., Waples & Do, 2010), suggesting that cumulative population declines may have to be very severe before they affect genetic estimates. Had our estimates of Ne been considerably smaller, or had we detected numerous subpopulations among which gene flow was restricted, then there may have been greater potential to document population size changes using genetic approaches. Given our results supporting a large, panmictic population, simulation studies are required to assess the sensitivity of coalescent-based estimates of Ne to population decline and to assess the utility of this approach for eastern red bats.

Our genetic data indicating panmixia and a lack of evidence for the use of different migratory pathways in different parts of the range highlights the need to consider the global implications of current and future fatalities associated with wind power. Despite growing conservation concern, current monitoring of bat fatalities at wind power developments is performed on an ad-hoc, site-by-site basis and may vary tremendously in scope according to local regulations. While such monitoring can provide valuable insights leading to site-level mitigation strategies or changes in turbine placement in some cases, biologists lack the necessary broader context within which to assess the long term, population-level impacts of observed fatality rates and management strategies at specific sites. For instance, site-specific, per-turbine thresholds to limit fatalities through curtailment (reducing turbine blade speed and operating time on low-wind nights in summer and fall to decrease fatalities; Baerwald et al., 2009; Arnett et al., 2011) ignore the fact that the demographic consequences of mortality extend well beyond any particular jurisdiction. Evidence from stable isotopes indicates that bats killed at wind power developments may originate from wide geographic areas (Voigt et al., 2012; Baerwald, Patterson & Barclay, 2014), and thus mortality at any given site can impact bat populations using geographically widespread catchment areas. Given that observed bat fatality rates at wind power facilities vary considerably among sites and regions (Arnett et al., 2008), our findings underscore the need for better data integration across jurisdictions and monitoring programs to adequately assess the cumulative demographic and genetic impacts of continued fatalities.

Supplemental Information

Table S1 Collection information for samples included in our study

Samples included in the primary dataset used in most analyses (samples collected between mainly between 2002 and 2006, but including samples from 1986 to 2007; dataset = 1) are listed first, sorted by population (in the same order as Table 1), followed by samples sequenced at HV2 but not included in site-level analyses. Samples included only in msvar analyses (2002 samples only, dataset = 2; 2010 samples only, dataset = 3) are subsequently listed, sorted by state and county. Note that samples included in dataset 1 may have also been included in dataset 2 (2002 only) if they were collected in 2002. Genbank accession numbers are provided for all unique sequences for HV2. For CHY, as a diploid nuclear marker, phased allelic sequences for each individual were submitted to Genbank, and so there are two accession numbers per individual sequenced. Sex (m, male; f, female), reproductive condition (RC: p, pregnant; l, lactating; pl, postlactating; s, scrotal; nr, non-reproductive, ns, non-scrotal), and age (a, adult; j, juvenile) are provided when available.

Click here for additional data file.

Table S2 Locus information for 16 microsatellites used to genotype L. borealis

Listed are load (combined PCR amplifications loaded on an ABI 3130 sequencer), PCR (multiplex panel and single-locus amplifications), annealing temperature (Ta), whether or not an extension is required (‘yes’ requires a final 30 min extension at 72 °C, ‘no’ skips this step), the number of cycles (we used a two-step PCR: the basic conditions started with 3 cycles, followed by 33 or 39 cycles [see main text]), locus, fluorophore, volume of each primer in PCR reaction (using GE Healthcare Illustra Pure-Taq Ready-to-Go Beads, we added 2 µl template, the listed volume of each primer, and water to a total volume of 25 µl), and the citation for the locus.

Click here for additional data file.

Table S3 Diversity of microsatellite loci, including number of alleles (NA), observed (HO) and expected heterozygosity (HE), estimated null allele frequency (F(Null)), and allelic richness (AR)

Click here for additional data file.

Table S4 Structure results for various numbers of clusters (K = 1–10) as produced by STRUCTURE HARVESTER

Mean LnP is the mean log likelihood for a given K across runs. Other terms are defined in Earl & vonHoldt (2012).

Click here for additional data file.

Table S5 Locus information for IMa2 analysis

Click here for additional data file.

Figure S1 Tukey boxplot of ancestral Ne from msvar analyses

Estimates are given on the log10 scale. Datasets A and B represent different subsamples of the full dataset from each respective year.

Click here for additional data file.

Figure S2 Tukey boxplot of the time of population size change from msvar analyses

Estimates are given on the log10 scale. Datasets A and B represent different subsamples of the full dataset from each respective year.

Click here for additional data file.

We are grateful to S Amelon, R Benedict, E Britzke, D Brown, C Butchkoski, T Carter, S Castleberry, MK Clark, S Darling, J Fiedler, L Finn, B French, E Gates, J Gore, M Gumbert, G Johnson, J Johnson, J Kiser, S Lambiase, G Libby, K Luzynski, A Miles, J O’Keefe, E Pannkuk, R Perry, L Pruitt, L Reddy, H Rice, L Robbins, D Saugey, M Schirmacher, J Schwierjohann, D Sparks, C Stihler, V Swier, A Tibbels, W Tidhar, C Willis, and L Winhold, and to the Angelo State Natural History Collection, the Tennessee State Rabies Testing Lab, and the Carnegie Museum of Natural History for providing tissue samples to support this research. J Glatz kindly provided GIS expertise to produce the range map. We also thank the Bats and Wind Energy Cooperative, and particularly E Arnett, C Hein, and W Frick for their input and support.

Additional Information and Declarations

Competing Interests

Author Contributions

Animal Ethics

Field Study Permissions

DNA Deposition

The authors declare there are no competing interests.

Maarten J. Vonhof and Amy L. Russell conceived and designed the experiments, performed the experiments, analyzed the data, contributed reagents/materials/analysis tools, wrote the paper, prepared figures and/or tables, reviewed drafts of the paper.

The following information was supplied relating to ethical approvals (i.e., approving body and any reference numbers):

The vast majority of samples were collected by other researchers as part of their biodiversity studies, and these researchers had to obtain appropriate permits and IACUC approval for their own research. Three samples were collected by one of the authors (MJV) under Institutional Animal Care and Use Committee approval (protocol 05-03-01).

The following information was supplied relating to field study approvals (i.e., approving body and any reference numbers):

Three of the samples collected by one author (MJV) were collected with permission from the Michigan Department of Natural Resources (permit SC1257).

The following information was supplied regarding the deposition of DNA sequences:

GenBank accession numbers KR337091–KR337257 and KR362302–KR362477.

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
