# Peer review of "Genetic approaches to the conservation of migratory bats: a study of the eastern red bat (Lasiurus borealis)"

_PeerJ, doi:10.7717/peerj.983_

## Round 0.1 · original submission · Major Revisions

· Academic Editor

Major Revisions

Based on the comments on two reviewers and in my own evaluation of the manuscript I recommend major revisions before the paper can be accepted for publication. This is not because the manuscript has any fatal flaws but because the revisions required are somehow extensive.

Apart from all the points highlighted by the reviewers it appears that the title, abstract opening and part of the introduction do not reflect the real content of the paper. This has also been mentioned by the reviewers, particularly R2. The work presents a population genetics analysis of bats, with particular emphasis on estimating effective population size and structuring, its justification being the difficulty of obtaining census estimates of the species. However, the experimental design, analyses, results and discussion do not bear any relation with the effect of the wind turbines and the study is not a case study for their effects on bats as claimed in the title. So, I recommend the authors to rephrase the title to make it reflective of the real content of the manuscript.

All of the comments of the reviewers should be addressed before the paper can be considered for publication. In particular, both reviewers expressed the need for clarification in the grouping for the analyses and the need of more detail in the description of the methods (e.g. likelihood in Structure and Flock). Please pay special attention to the comments of R1 with regards to sex-biased dispersal. R1 also recommends editing parts where sentences are too wordy and difficult to follow. Finally, I agree with R2 that a relationship between Ne and turbines is not demonstrated in the ms.

Reviewer 1 ·

Basic reporting

The manuscript adheres to policies and for the most part is clear and conforms to professional standards. In a few instances some sentences read as long and wordy and careful editing of those that contain multiple ideas would improve clarity. A few references are missing which would add to breadth of the study but nothing major is missing. These references and a few examples of wordy sentences have been included in the specific comments on the manuscript. The article fits the standard sections of the journal submission structure.

Experimental design

The research question is very broadly defined and further refinement is needed to clarify the how the types of analyses performed relate to specific questions/aspects of the study. These are better developed in the abstract and should be incorporated into the main body of the manuscript. The experimental design is sufficient to address the overall goal of the study and no major changes to analyses are required. A few added details in the methods would clarify some minor points. An alteration to the structure of how the methods are presented would facilitate the reader in better following what lab techniques/analyses were performed for each objective of the study. This is further outlined in the specific comments.

Validity of the findings

The data are of generally of sufficient quantity and quality to address the research question. It is not clear if the data will be made available in a repository and this needs to be addressed. A few results of the population genetic analyses should be expanded on (see specific comments). The results of the effective population size are analyzed appropriately given the limitations of the data, and general knowledge of the species. Conclusions based on these data are supported with the limitations adequately discussed to bound their inferences appropriately. Assessment of sex biased dispersal using the data from this study needs to be addressed as the results do not support the conclusions as stated - See specific comments for more information.

Comments for the author

There are many instances where sentence structure is complex and wordy. Several sentences would benefit from editing where they contain multiple ideas.

Examples:
Abstract, last sentence beginning with “Our results…” As written this sentence doesn’t make sense and is rather long winded.

Introduction, paragraph 2, sentence beginning with “ While estimates…” End this perhaps after CMR methodology where the last phrase would be a stand-alone sentence that shows the importance of the message of this phrase.

Introduction, paragraph 4, sentence beginning with first sentence “While it is not possible”. End this after size (Ne).

Introduction, paragraph 5, sentence beginning with “Although estimates of”. End this after “next generation)”. – side note, excellent to bring this idea up directly as many readers may not be familiar with the concept of Ne or the differences between Ne and NC

Specific comments

Abstract – “large dataset” large is an ambiguous term. There are many datasets of fish populations genetics in the 1000’s, delete as it adds nothing.

Introduction, paragraph 3, 3rd sentence beginning with “In addition,”. Often sex is very informative and necessary depending on the inferences trying to be made with genetic data given a specific marker. Temper sentence as it is more than just “useful”.

Introduction, paragraph 3, sentence beginning with “Molecular markers”. Up to this point, this paragraph discusses generalities of genetic approaches and what they can tell us about populations. Thus I would use a more general reference here, or if this is to be tailored to specifically about bats then use a couple of references to show how these approaches have been useful overall to bat studies.

Introduction, paragraph 4, last sentence. Simplify “And provides us with” to “thus providing”.
The goal of this work is defined too broadly. I think it would benefit from having some explicit objectives laid out that would provide the framework for the methodologies used - the way it is laid out in the abstract should be incorporated into the ms body. This could include some predictions on what you expected to find with each given the other literature on bats (European studies) with a similar migratory life history (N. noctula, P. pipistrellus; Petit et al 1999,Petit and Mayer 1999, Bryja et al 2009)

Methods.

Based on the map I would say your samples come from the core of the range and are concentrated in the east rather than representing the entire range. With increased sampling of bat biodiversity (sadly from wind farms or oil and gas exploration) new data are showing that red bats may be further west/north than previously thought. E.g. Nagorsen and Paterson 2012 (BC, Canada) and detection of calls in southern AB near the Rockies (Baerwald and Barclay 2009) or northern AB (Canada, Grindal et al 2011).
Also, the map shows an open circle in the NW portion of the range perhaps on the border of AB and SK (Canada) but this information is missing from your Table S1.

Methods. At first the datasets are split out (nicely) to indicate samples sizes and descriptions for each set. Then the laboratory methods are presented as genotyping, HV2 sequencing, CHY sequencing. In the next section, it jumps to analysis of population structure starting with mtDNA. This jumps around a bit for the reader making it hard to follow. I suggest sticking with the structure of the 3 datasets and referring back to each as a higher level of organization for this section with little “signposts” to the reader. For example, in describing testing for differences among sampling locations (or groups of sampling locations) in allelic richness, FIS and HE, it is not overly clear what dataset this was performed on.

Methods – Ima2. Is this standard to edit the sequence data to an infinite site model?

Results – Are there plans to have the haplotypes or genotypes definitions placed in a repository? This may be valuable to make available for other future studies.
Results – a pattern of high haplotype diversity and low nucleotide diversity suggestive of population growth has been shown other bat species such as N. noctula (Petit et al. 1999) and M. lucifugus (Burns et al. 2014).

Results – second paragraph. I agree in principle with running the analysis without the 4 loci that are not in HWE in a preliminary run to check that the analyses are concordant with analyses that have them included. However, I think a table should be added to the supplementary information that shows the He, Ho, P-value for each locus so that the reader has some information to assess this rather than a rather short sentence about it. Did you check for the presence of null alleles in these four loci? If so, these results should be included and perhaps a sentence or two on why this is or is not an issue with the loci should be added. If not tested for null alleles, then these should be performed and included. This is especially important since some of these loci were developed for other species and some appear to be specific to red bats but have not been previously published. This information will help to inform other researchers going forward, especially if your study is used as a baseline.

Results- third paragraph, the log likelihood value that supports K = 1 for STRUCTURE results should be reported since no other figures or info related to this analysis are included. Same thing for the FLOCK analysis, report the LLOD. Results of AMOVA should include the P value for the FST results reported and in the methods section for this, include the method used to assess significance.

Discussion
The idea of migratory pathways and distinct genetic populations is somewhat buried in the introduction. Since it is not brought up in the “goal’ of the study, I think it would be worthwhile to clearly discuss how your genetic results supports this notion (although see above comments on fleshing out your goal/objectives more).

Second paragraph. In the final thoughts of this paragraph you discuss the idea of female dispersal. In mammals the pattern is generally male biased although the degree of this can vary, and some exceptions are known – including in some tropical bats (Nagy et al 2007, Nagy et al 2013). The AMOVA results from mtDNA showed “low levels of differentiation” with an FST=0.0113. The AMOVA results from nuclear DNA showed an FST = 0.0044. This finding of higher structuring on mtDNA compared with nucDNA is typically interpreted as a male bias in dispersal not a female bias. The lack of strong structuring on mitochondrial DNA may suggest that the male bias is somewhat reduced in this species, or perhaps is closer to parity between the two, but it may be premature to suggest gene flow takes place via female movements and mating. An explanation of why a female bias is thought to exist for red bats is needed since the mtDNA to nucDNA FST results suggests otherwise and other, more formal methods are not used in this study that would perhaps better address the question of sex biased dispersal (see Petit et al 2001).

Third paragraph. Nice synthesis of the different analysis of Ne and concordance of results.

Recent work using stable isotopes has shown that wind farms can have large catchment areas for the origins of the bats that are killed there (Voigt et al 2012, Baerwald et al 2014,). Incorporation of these studies would likely be informative to your discussion of the far reaching consequences of fatalities from wind energy developments.

Reviewer 2 ·

Basic reporting

no comments

Experimental design

The methods are not sufficiently reported to be reproducible. Please see my comments to authors.

Validity of the findings

no comments

Comments for the author

Peer J Genetic approaches to understanding the population-level impact of wind energy development on migratory bats: a case study of the eastern red bat (Lasiurus borealis).

Overall this paper seeks to examine population differentiation and effective population size of a broad ranging arboreal bat. This species (Lasiurus borealis) has been identified as one of the top three species killed by wind turbines. The concern is that this has or will affect their population numbers. Because it is difficult to get census numbers on bats for many reasons the authors chose quantitative approaches for estimating the number of individuals contributing to gene flow as a proxy. Although the authors have collected samples from across the geographic range and have decent samples sizes the way they divided datasets and applied various analyses are unclear. I would say revision with a mind to clarity in the methods is critical to the success of this manuscript. Further providing solid reasoning of how this study is connected to understanding the effects of wind turbine mortalities is also critical. I would suggest using other systems where a broad ranging species was hurt by a disease (e.g. White nose syndrome and little brown bats) or an anthropogenic activity as a means to demonstrate how knowledge of baseline population data can help our understanding or help the management of the problem. I see that stated hear but no evidence is provided to support the claim. Therefore the link between the work and the problem is weak. Please see detailed comments below:
Line and page numbers would have helped make the review more efficient.
Introduction:
First line remove “the” after “burning”
Last sentence: how will this study provide “context for understanding the population-level impacts of mortalities due to wind power”. Convince me of the connection between the study and wind power impacts.
Methods:
Page 7, 1st pgh change “one state” to “a single”. Why did you generate three data sets. Some justification is necessary. You have some about the sites where you have enough samples to call a population for population genetic analyses but there are more data sets than that and it is unclear why your sampling is subsampled to generate many data sets.
2nd pgh maybe need to divide the data sets under here (n=3) into 1a), 1b), and 1c) so that later in the paper each can be referred to separately as you actually do different things to them. After “2000-2006” add “(N=284 or whatever the right number is please)”. Is the Ne estimator referred to here different than the one described in #2? I think these need to be defined and have appropriate references. I know you get into the details later but it is confusing here without that you define Ne as a single timepoint in #2 but don’t give any indication in #1 about what kind of estimator it is. I think just adding a reference after each one could be a simple solution. After “(N>15) for” add “each of”. After “12 localities” add “(population samples)” or whatever you will call these. You seem to use three different terms including “population samples”, “localities”, and “sites”. Pick one, define it, and stick with it to improve clarity. After “bats were captured” add “either”. Before “A 408 bp fragment” add “Further,” How did you determine an area to be an appropriate site to call it a population sample? Was this based on biology of the species (e.g. home range)? Why were not all locations or individuals sequenced? How was this decided? Add “nuclear” after “651 bp fragment of the”. Why did you create a subset for the nuclear gene. The methods really suffer from a lack of clarity. There is a lot going on here with many datasets being analyzed in different ways and without the clarity the reader gets lost and loses interest and faith in the outcome. Providing sufficient details will help alleviate this issue.
Need a map that shows datasets from 2) and 3) so that the reader can see how well the geographic range was subsampled.
Under 3) were there no samples from first dataset included here? You made this clear for 2) but not for 3). Define Ne estimators or use “(see below)”.
Laboratory methods:
Perhaps providing the multiplexes in a table in supplementary information or on data dryad could help researchers replicate your work more easily than having to track you down.
2nd pgh delete “of the 408 bp fragment”.
3rd pgh exchange “chymase” for “CHY”. Were some clean sequences generated here without cloning? Unclear methods.
Last pgh again this is unclear. Were your final 103 sequences from individuals or phased alleles? Here you say 36 individuals, so did most individuals (N= 103-36) provide a single nuclear sequence?
Analysis of Population Structure:
1st pgh move this to after microsatellite analysis methods as in the previous section you addressed microsatellite methods first. Did you do this for all 295 individuals or the 218? Throughout this section it is confusing about which data set gets which analyses maybe use labels in parentheses (e.g. 1b and 1c).
2nd pgh Is this all the 1a dataset? Add “microsat” between “each” and “locus” in the first sentence. Make it clear which data sets and which markers are used for each analytical method. I think if you standardize what you are calling a “site” or “location” or “population” then you could use that instead of “sampling locations (or groups of locations)” here is an example of where it gets confusing because terminology keeps changing and because there is a lack of defining what you grouped or how you grouped or why you grouped.
3rd pgh Which datasets were analyzed with these methods? Would be good to have a reference for the last sentence of this pgh.
4th pgh how are you sure that 20 iterations leads to genetically homogeneous clusters?
5th pgh which data set were these analyses performed on?
Estimation of Ne:
1st pgh, 2nd sentence why couldn’t you do a temporal estimation with your data? Why did you perform three analyses and could you compare and contrast them to allow the reader to understand whyyou would pick three and not just one?
Msvar:
1st pgh I have a concern over the subsampling of 100 individuals. What kind of representation of the geographical range doe this subsampling have and should there be concern over bias if the geographical range is not well subsampled.
2nd pgh after “Lamarc” add “(see below):
IMa2:
1st pgh didn’t you also estimate mutation rates? Why did you not subsamples twice like you did for msvar analyses? Are the sample sizes the same for the microsat and mitochondrial DNA datasets? IF not is this a problem? How did you edit sequence data to fit an infinite sites model?
Lamarc:
Was this only sequence data? What was the sample size for each locus CHY=103? And HV2=295? Is there a problem that they are different samples sizes?
Results:
1st pgh I think this should be moved to after the microsatellite reporting to reflect methods. Doesn’t high haplotype diversity and low nucleotide diversity also suggest loss of haplotypes from a previously large and genetically diverse population. Mismatch distribution would help visualize and test this? I think there are other seminal population genetic references to go along with the Russell et al paper.
2nd and 3rd pghs to which data sets do these results apply?
Ne estimation:
Were the markers with differential inheritance really expected to give complementary evidence? There are inherently different effective population sizes of mtDNA and nDNA.
Msvar:
What are these A1-A3 and B1-B3? Previously you only mention A and B. You say here that there was no consistent difference between subsamples and the full dataset but there was nothing in the methods about comparing subsets to the full data set. How was this done?
Discussion:
1st pgh, 2nd sentence change “Furthermore” to “Subsequently”.
3rd sentence what about Piaggio and Perkins 2005?
2nd pgh, 1st sentence what about Piaggio, Navo, and Stihler 2009?
3rd pgh, 3rd sentence what about inheritance modes and their contribution to different results?
4th pgh I would take out this pgh as it is distracting and weakens the argument for using Ne in the context you are arguing to use it in.
5th pgh, 2nd sentence you need a reference here. Actually the perfect sentence for here is in the next pgh where you cite Kunz et al. 2007. Move it here?
6th pgh I am not convinced of the connection between Ne and the management of wind power fatalities. This concluding paragraph needs to be strengthened.

---

## Round 0.2 · Minor Revisions

· Academic Editor

Minor Revisions

Both reviewers consider, and I agree with them, that the reviewed manuscript has been considerably improved. The authors have made a very good job in addressing all of the comments and the reviewers have now only a few minor comments left to be addressed. Please pay particular attention to the comments of Reviewer 1 about the missing references and the map and consider the suggestions from Reviewer 2. I am a bit surprised that the sequences have not been submitted to Genbank yet, mainly considering that an embargo period is possible to ensure that they are not made public before the article is published.

Reviewer 1 ·

Basic reporting

N/A

Experimental design

Incorporation of the objectives from the abstract into the main body has strengthened the presentation of the research question. A fleshing out of a few more of the details in the methods section, including adding in information to supplemental material, has increased the utility of this paper as a baseline for future work.

Validity of the findings

As per the previous review, the authors did a nice job of describing population genetic structure and effective population size for a species of high conservation concern. I believe they provided and discussed appropriately, the context of how this work relates to the specific conservation issue of wind farms.

Comments for the author

The manuscript has made substantial progress. It now reads well throughout, including in the methods section which has now been reorganized. I recommend this article should be accepted and offer only a few minor comments below.

In lines 416 & 417 of the discussion (sentence on previous phylogeographic studies), perhaps include the references in line 414 when they are first referred to, rather than in the second sentence

The references in line 511 are missing from the references section at the end. Baerwald et al., 2014 and Voight, et al., 2012.

The map in Figure 1 has a strange "extra" island associated with eastern Canada. It is just off the island of Newfoundland to the southeast and has a very linear geography - perhaps it is associated with a marine zone of some sort?

Reviewer 2 ·

Basic reporting

No comments

Experimental design

No comments

Validity of the findings

No comments

Comments for the author

The authors have addressed the reviewer’s comments and concerns. I particularly appreciate the change in title and the focus on clarifying the methods. I never doubted the methodological approach for utilizing different datasets I just was never clear on the how or the why of the allocation of data amongst the analytical approaches. The revised manuscript addresses the confusion and is succinct in the description of why datasets were divided and how that was accomplished.

Further, I want to be clear that I never doubted the contribution of this work to the body of knowledge required to begin to build appropriate management strategies for the effects of wind power on this species. Rather, I felt the argument, as presented, was unconvincing. The revisions have corrected this issue in my eyes and strengthened the connection between the work and our need to manage the effects of wind power fatalities for this species. We now know our management units are quite large and that it will be difficult to see an effect immediately but that does not alleviate the possibility that it might be a significant impact on these populations before we can ever detect it through genetic methods. I appreciate the effort the authors put into addressing my comments even though they did not appreciate my perspective.

Minor comments:

Lines 107-112- nice connection, thank you.
Lines 112-129 nice clarification, thank you.
Line 132 remove first “of”
Lines 134-140 Much clearer now, thank you.
Lines 142-145 this provides clarity but seems to be misplaced. How about moving to line 206?
Line 174 and Lines 188-189 why use two different clean up approaches? Were these done in different labs? It may not be critical to state but it did make me wonder…
Line 241 are pre-defined sites the 12 from line 129?
Line 242 and lines 249-250 I think these sentences are repetitive.
Line 269 perhaps add “(see results)” after “red bat population”.
Line 385 estimates from HV2?

---

## Round 0.3 · accepted · Accept

· Academic Editor

Accept

Dear Dr Vonhof,

I am very pleased to let you know that after the revisions your paper can be accepted for publication. Congratulations on a very interesting study. I only have a last comment as have just noticed the following note from the Editorial office:

Please be aware that the authors state this a case study in the title, ((which PeerJ doesnt accept), but it doesn't appear to be truly a case study so the author would be advised to change the title for clarity's sake. Could you ensure that you change the title accordingly?